# Structures of the archaerhodopsin-3 transporter reveal that disordering of internal water networks underpins receptor sensitization

Juan F. Bada Juarez [1,9], Peter J. Judge [1,9], Suliman Adam [2], Danny Axford [3], Javier Vinals[1], James Birch [3,4], Tristan O. C. Kwan [4,5], Kin Kuan Hoi [6], Hsin-Yung Yen[7], Anthony Vial [8], Pierre-Emmanuel Milhiet [8], Carol V. Robinson [6], Igor Schapiro [2], Isabel Moraes [4,5✉] & Anthony Watts [1✉]

Many transmembrane receptors have a desensitized state, in which they are unable to respond to external stimuli. The family of microbial rhodopsin proteins includes one such group of receptors, whose inactive or dark-adapted (DA) state is established in the prolonged absence of light. Here, we present high-resolution crystal structures of the ground (light-adapted) and DA states of Archaerhodopsin-3 (AR3), solved to 1.1 Å and 1.3 Å resolution respectively. We observe significant differences between the two states in the dynamics of water molecules that are coupled via H-bonds to the retinal Schiff Base. Supporting QM/MM calculations reveal how the DA state permits a thermodynamic equilibrium between retinal isomers to be established, and how this same change is prevented in the ground state in the absence of light. We suggest that the different arrangement of internal water networks in AR3 is responsible for the faster photocycle kinetics compared to homologs.

[1] Biochemistry Department, Oxford University, South Parks Road, Oxford OX1 3QU, UK. [2] Fritz Haber Center for Molecular Dynamics Research, Institute of Chemistry, Hebrew University of Jerusalem, Jerusalem 9190401, Israel. [3] Diamond Light Source, Harwell Science and Innovation Campus, Didcot OX11 0DE, UK. [4] Research Complex at Harwell, Rutherford Appleton Laboratory, Harwell Science and Innovation Campus, Didcot OX11 0FA, UK. [5] National Physical Laboratory, Hampton Road, Teddington, London TW11 0LW, UK. [6] Chemistry Research Laboratory, Oxford University, Mansfield Road, Oxford OX1 3TA, UK. [7] OMass Therapeutics, The Schrodinger Building, Oxford Science Park, Oxford OX4 4GE, UK. [8] Centre de Biochimie Structurale (CBS), INSERM, CNRS, University of Montpellier, Montpellier, France. [9] These authors contributed equally: Juan F. Bada Juarez, Peter J. Judge. ✉email: isabel.moraes@npl.co.uk; anthony.watts@bioch.ox.ac.uk

Transmembrane (TM) receptor proteins are ubiquitous in biology. They enable cells to sense and to respond to their environment by undergoing conformational changes on ligand binding or light absorption. In addition to their active and resting states, several receptor proteins have a desensitized or inactive form, in which their responsiveness to external stimuli is reduced. Desensitization results in the uncoupling of the receptor from its downstream effectors, thus reducing the magnitude of the cellular response.

Desensitization is commonly achieved through reversible covalent or non-covalent modifications, which typically modulate intramolecular bonding networks, to stabilize a conformation which is distinct from the active resting or ground state of the receptor[1]. In the case of some G-protein coupled receptors (GPCRs), including rhodopsin and the $\beta_2$ adrenergic receptor[2], a Ser/Thr kinase phosphorylates residues on the cytosolic face of the protein, which promotes the recruitment of an arrestin that in turn blocks the binding site for G$\alpha$[3]. Other GPCR desensitization mechanisms involve changes in glycosylation patterns (e.g., for the dopamine D3 receptor[4]) and fatty acid conjugation (e.g., for the vasopressin receptor V2R[5]). In the case of ion channels, desensitization may occur in the persistent presence of a ligand[6] and may be achieved through a relatively subtle conformational change to produce a closed state, which is distinct from the resting state[7–9].

Archaerhodopsin-3 (AR3, from the archaebacterium *Halorubrum sodomense*) is a photoreceptor which, like all eukaryotic GPCRs, has seven TM helices. Although the wild-type protein is more usually classified as a light-driven proton pump, mutants of AR3 are commonly used in optogenetics experiments to enable individual neurons to be stimulated or silenced (by altering the permeability of the cell membranes to cations when illuminated at specific wavelengths) or as membrane voltage sensors[10–13]. AR3 is particularly suitable for these applications, since the protein has been suggested to have faster photocycle kinetics than many of its homologs[13] (including bacteriorhodopsin (bR) from *Halobacterium salinarum*), although the current produced by recombinant AR3 expressed in *Xenopus* oocytes has been measured as comparable to that of bR[14]. Development of AR3 mutants (commonly termed Arch in the optogenetics field) has been hampered both by the absence of high-resolution structural information[15–18] and by a lack of understanding of the mechanisms of receptor desensitization[19–22].

Although it is not a direct homolog of the Class A GPCR rhodopsin, AR3 undergoes a similar, highly ordered sequence of conformational changes (known collectively as the photocycle) after the ground state of the protein is stimulated by a photon of appropriate wavelength. To enable them to absorb light, the immature forms of AR3 and rhodopsin are modified after translation by the addition of retinal, which is covalently conjugated to a lysine sidechain via a Schiff Base (SB) linkage to produce the retinylidene chromophore. It is a light-induced change in the isomerization state of the retinal, detected by the surrounding receptor, that initiates the progression around the photocycle[23–25].

The desensitized form of archaeal, retinal-containing photoreceptors (the so-called dark-adapted (DA) state) is established in the absence of light and is characterized by a thermal equilibrium between at least two conformations, one with all-*trans* retinal (in common with the ground state) and the other with 13-*cis* 15-*syn* retinal. (Many of the early photocycle intermediates contain 13-*cis* 15-*anti* retinal[26–28].) To date, it has been unclear how the change in chromophore isomerization occurs with no apparent input of energy, to produce the DA state from the ground state or light-adapted (LA) state. It is also not clear why a thermal equilibrium between *cis* and *trans* retinal is established in the DA form, but only all-*trans* retinal is found in the LA state[26–29].

In this work, we have determined the crystal structures of the DA and LA states of the wild-type AR3 photoreceptor, solved to 1.3 and 1.1 Å, respectively. Our structures reveal the different conformations of the chromophore and the internal hydrogen-bonding networks that control the molecular mechanism of receptor desensitization and resensitization. Using quantum mechanical/molecular mechanical (QM/MM) approaches, our study explores the differences between the two states in both the activation energy barrier for conversion between *cis* and *trans* retinal, and also the equilibrium position between the two isomers. We propose that the apparent re-arrangement of internal water networks in AR3 may be a more general feature of receptor activation and deactivation beyond the family of archaeal photoreceptors.

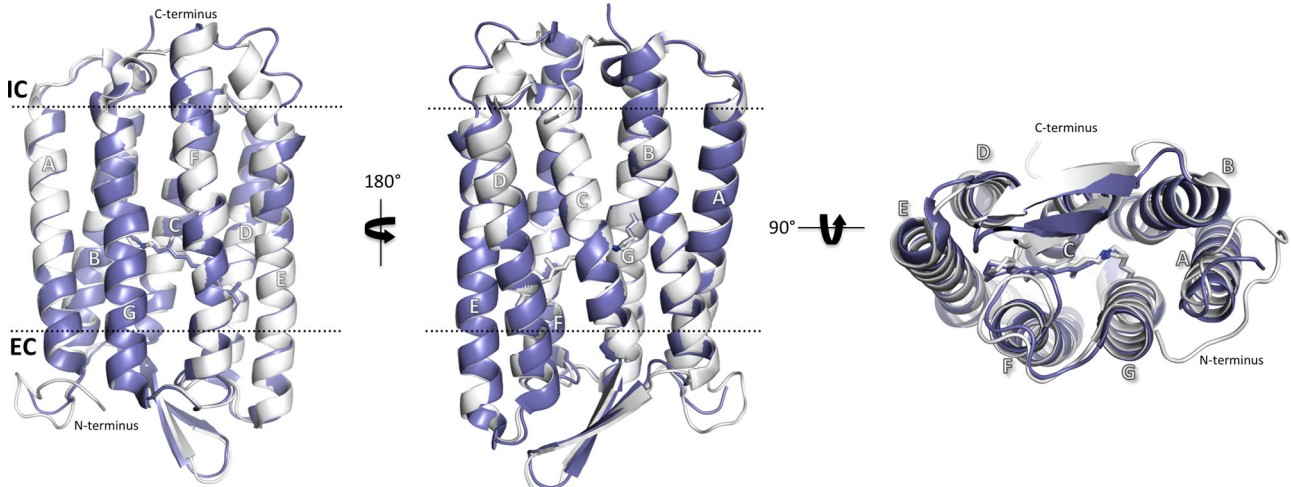

**Fig. 1 Comparison of the light-adapted (LA) AR3 (6S6C [http://doi.org/10.2210/pdb6S6C/pdb], white) and bR (5ZIM [http://doi.org/10.2210/pdb5ZIM/pdb] purple) crystal structures.** The approximate positions of the extracellular (EC) and intracellular (IC) membrane interfaces are shown as black dotted lines. The retinylidene chromophore (formed by the post-translational conjugation of retinal to a lysine sidechain) is shown in stick representation. The transmembrane helices (shown in ribbon representation) are labeled from A to G. The N termini of both proteins face the extracellular (EC) side of the membrane and the C-termini face the intracellular (IC) side.

## Results

**Crystal structures of the DA and LA states of AR3 solved at atomic resolution.** Wild-type AR3 was purified from *H. sodomense* cells that had not been genetically modified and protein crystals were grown in lipidic cubic phase (LCP) (Supplementary Fig. 1)[30,31]. Both purification and crystallization of AR3 were performed in the absence of detergents ("Methods"). The LA ground-state structure of AR3 was solved from crystals which had been illuminated under a white tungsten light for two minutes prior to *cryo*-freezing. Crystals that had not been exposed to light were used to determine the structure of DA AR3. Diffraction data (Supplementary Fig. 1) from different crystals were merged to obtain the final data sets. Both structures were solved by molecular replacement using the coordinates of Archaerhodopsin-1 (AR1, 1UAZ [https://doi.org/10.2210/pdb1UAZ/pdb])[32] to a resolution of 1.3 and 1.1 Å for the DA and LA states respectively (Supplementary Table 1).

Similar to its homologs (including bR, with which it shares 59% sequence identity, Supplementary Fig. 2), AR3 has seven TM helices and a single, extracellular-facing, two-stranded β-sheet (Fig. 1), consistent with circular dichroism data (Supplementary Fig. 3a). In addition, retinal is covalently bound, via a SB to residue Lys226, thus creating the retinylidene chromophore. Ultraviolet-visible spectra suggest that AR3-rich membranes also contain a bacterioruberin pigment (Supplementary Fig. 3b); however, unlike in the case of Archaerhodopsin-2 (AR2)[33], this molecule could not be resolved from the residual electron density during refinement. Several lipid fragments and a large number of water molecules are also observed across the structures. Although crystallographic data shows one molecule in the asymmetric unit, atomic force microscopy images of patches of the claret membrane from wild-type *H. sodomense* cells (Supplementary Fig. 4) suggest that, like bR, AR3 is trimeric and forms a hexagonal lattice in the native organism[34].

**Retinal is resolved in two conformations in each state.** A striking aspect of the two high-resolution *cryo*-temperature structures is that the retinal conformations are very well resolved in both cases (Fig. 2 and Supplementary Fig. 5a, d). When all-*trans* or 13-*cis* retinal isomers are fitted at 100% occupancy for the DA and LA structures, strong positive and negative peaks from the calculated m$F_{obs}$–D$F_{cal}$ maps are observed around the SB lysine and the β-ionone ring, suggesting a second retinal conformation (Supplementary Fig. 5) in each state.

In the DA state (6GUX [https://doi.org/10.2210/pdb6GUX/pdb]), 13-*cis* and all-*trans* retinal isomers are modeled into the electron density in a calculated occupancy ratio of 70% and 30%, respectively (Fig. 2, left). The isomer ratio observed in the AR3 DA structure is therefore similar to that previously reported from liquid chromatography studies for the DA state of other photoreceptor homologs (67%:33%)[35]. Although the positions of the carbon atoms nearest the SB (C12, C13, C14, C20) show the largest displacement when the isomers are compared, there are no significant differences in the positions of the SB nitrogen atom itself or the remainder of the Lys226 sidechain.

The conformations of the two retinal isomers in the 6GUX [https://doi.org/10.2210/pdb6GUX/pdb] structure were subsequently optimized using a QM/MM approach. The calculated structure of the *cis* isomer was in excellent agreement with the coordinates of the crystal structure. The calculated *trans* structure has a reduced twist along the main axis of the chromophore, when compared to the crystal structure, with the greatest discrepancies observed for the C10-C13 atoms in the retinal backbone as well as the C20 methyl group (Supplementary Fig. 6).

In the LA ground-state structure (6S6C [https://doi.org/10.2210/pdb6S6C/pdb]), retinal is resolved as two different conformations of the same all-*trans* isomer (Fig. 2, right), with relative occupancies of 75% and 25%. In this state, the greatest differences in atom position are observed in the Lys226 side chain (specifically Cδ) and the SB nitrogen atom. The calculated structures show an even larger difference between the two forms and suggest both a reduction in the axial twisting in the major conformation (trans1, Supplementary Fig. 6) and an increase in the twist in the minor conformation (trans2, Supplementary Fig. 6). The crystal and calculated structures are however in agreement that the significant difference in the position of the SB N atom is sufficient to cause a break in the H-bond to internal water molecule W402 (Supplementary Fig. 6).

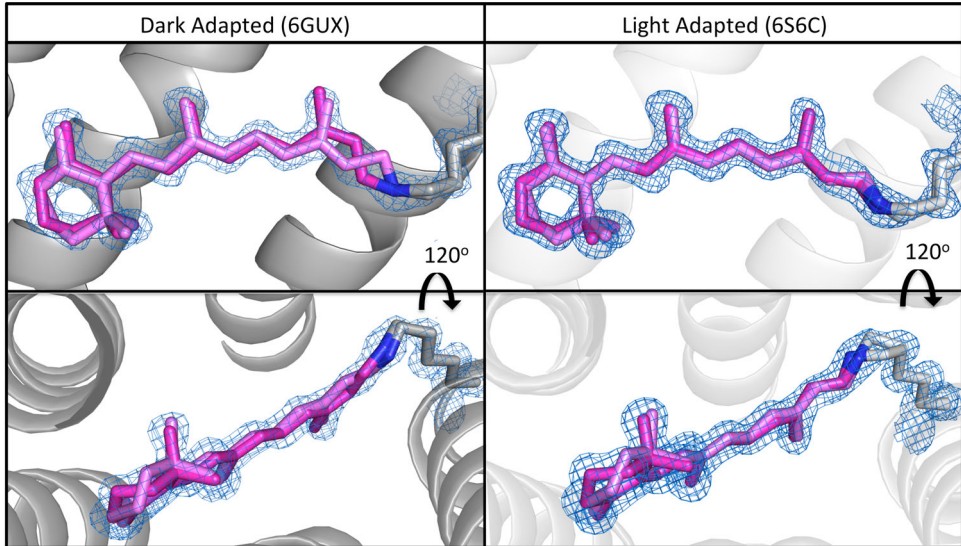

| Dark Adapted (6GUX) | Light Adapted (6S6C) |
| --- | --- |

**Fig. 2 Comparison of the conformations of retinal in the DA (6GUX [http://doi.org/10.2210/pdb6GUX/pdb]) and LA (6S6C [http://doi.org/10.2210/pdb6S6C/pdb]) states of AR3.** In the DA state (left) the C13 = C14 retinal bond has been modeled with 70% *cis* and 30% *trans* isomers (colored in dark and light pink, respectively). In the LA state (right) retinal (colored in dark and light pink respectively) is modeled in the all-*trans* state only, but as two different conformers. Movement of the β-ionone ring is also observed in both structures. The 2$F_{obs}$–$F_{calc}$ electron density maps (blue mesh) around the retinal and Lys226 are contoured at 1.2σ.

Two variants of the puckering of the β-ionone ring are also observed in each crystal structure, characterized by a movement of the C3 atom of ~1.1 Å perpendicular to the plane of the ring (Supplementary Fig. 5a–f, left). The distance from C3 to Ser151 in the DA state is 3.4 Å for the *cis* form and 4.4 Å for the *trans* isomer. The changes in this part of the chromophore are also visible in the structures calculated by QM/MM (Supplementary Fig. 6). Nuclear magnetic resonance studies of inactive rhodopsin in the dark state also suggest that there is disorder around the C3 position of the β-ionone ring[36]. We therefore interpret this ring puckering (also observed in the calculated retinal structures, Supplementary Fig. 6) to indicate that this part of the chromophore is less constrained by the protein environment and is more disordered than has been previously reported in lower-resolution structures of homologs.

With the exception of the atoms in the β-ionone ring, the carbon skeleton (C6 to C15 inclusive) of the all-*trans* retinal in the DA state crystal structure, matches exactly the major conformation of the chromophore in the LA state. (There are significant differences in the position of the C atoms nearest the SB in the second conformation of all-*trans* retinal in the LA state.) There would therefore appear to be at least two independent mechanisms for receptor sensitization. In the first mechanism, the population of DA receptors with all-*trans* retinal may be directly stimulated by light to enter the photocycle. As this DA-*trans* population is depleted, the thermal equilibrium between the isomers acts to convert the 13-*cis* isomer and under continuous illumination produces a population of active LA AR3 proteins. A second mechanism is also proposed, in which the 13-*cis* population is capable of being excited under illumination to produce a 13-*trans*,15-*syn* chromophore, which then rapidly relaxes to the 13-*trans*,15-*anti* isomer observed in the ground or resting state. It would be possible for the two mechanisms to occur simultaneously and in parallel.

**The SB electronic environment determines the relative stability of retinal isomers.** The stability of the 13-*cis* and all-*trans* retinal in the two structures was compared by calculating the potential of mean force (PMF) for the rotation about the C13 = C14 double bond for all conformers (Fig. 3 and Supplementary Fig. 7). For DA AR3, the 13-*cis* isomer is energetically more favorable than the all-*trans* isomer ($\Delta G_{cis\text{-}trans} = -1.9$ kcal/mol), which is in qualitative agreement with the presence of the two isomers in the deposited crystal structure (with the *cis* isomer being the dominant species). In the case of LA AR3, the all-*trans* isomer is more stable, ($\Delta G_{cis\text{-}trans} = 10.9$ kcal/mol). The larger energy difference means that the 13-*cis* isomer is far less likely to be observed than the all-*trans* isomer in the LA state, consistent with the crystal electron density maps in which only all-*trans* retinal can be detected (Fig. 2). The activation energy for interconversion between the retinal isomers is calculated to be 4.4 kcal/mol higher for the LA form (21.5 kcal/mol) compared to the DA form (17.1 kcal/mol). These calculations are consistent with experimental observations of other microbial rhodopsins that show that, in the DA form, both retinal isomers can be found at room temperature, consistent with a thermal isomerization in the absence of light ($kT_{298K} = 0.59$ kcal/mol); however, in the LA form, absorption of a photon is required to convert between the two isomers[37,38].

Finally, although the position of the SB nitrogen atom of retinal appears to change little between the *cis* and *trans* forms in the DA structure (6GUX [https://doi.org/10.2210/pdb6GUX/pdb], Supplementary Fig. 5b,c), a subtle movement of 0.5 Å in the position of the same atom in the LA structure (6S6C [https://doi.org/10.2210/pdb6S6C/pdb]) is observed (Supplementary Fig. 5e,f). Similarly, the Lys226 side chain is resolved in only one position in

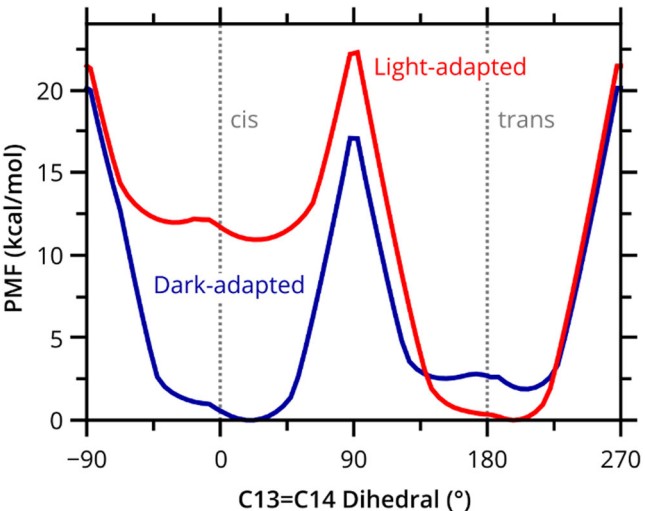

**Fig. 3 Calculated potentials of mean force (PMF) for the isomerization of the C12–C13 = C14–C15 dihedral of retinal in DA (blue) and LA (red) AR3.** The PMF was computed by sampling the retinal isomerization from all-*trans* to 13-*cis* and vice versa. Each point on the curve is generated from two independent 0.5 ns QM(SCC-DFTB)/MM MD trajectories, initiated from two separated equilibrated starting structures. The protein backbone was fixed in place; however, all other atoms (including those in the chromophore and amino acid sidechains) were allowed to move.

the DA structure (Fig. 2 left and Supplementary Fig. 5a–c), whereas in the LA structure this side chain is resolved in two conformations (see Fig. 2 right and Supplementary Fig. 5d–f). In the LA state, these sidechain movements allow the SB nitrogen atom to approach to within 2.9 Å of the Thr99 side chain and may support an H-bond at this position, which is not achievable in the DA state (Supplementary Table 4). Since there is no significant change in the position of the hydrophobic residues that line the retinal binding pocket (including Tyr93, Trp96, Leu103, Met155, Trp192, and Trp199), it would appear that the differences in the relative stabilities of the *cis* and *trans* retinal isomers in the DA and LA states are solely dependent on the different electronic environments around the SB[39].

**The SB N atom position influences internal water dynamics.** Light-driven proton pumps such as AR3 rely on a precisely coordinated network of internal water molecules to mediate proton translocation across membranes[37,38]. The movement of internal water molecules in bovine rhodopsin has previously been shown, using MD simulations, to respond to changes in chromophore conformation[40]. As in bR, both the LA and the DA structures of AR3 reveal a "quasi-planar" pentagonal hydrogen-bonding network in the SB region, formed by three internal water molecules (W402, W401, and W406) and two negatively charged sidechains (Asp95 and Asp222) (Fig. 4a, b and Supplementary Fig. 8)[41–45]. This region is known to undergo structural changes during the early stages of the microbial rhodopsin photocycle and plays a key role in the transfer of an H[+] ion from the protonated SB to the proton release complex (PRC)[41–45]. By comparing the occupancies and static positions of water molecules and amino acid side chains resolved in the two *cryo* crystal structures, we can infer information about their likely dynamics at physiological temperatures.

Close inspection of the structures reveals the coupling between the chromophore and the pentagonal H-bond network. The greater ambiguity in the position of the retinal SB N atom in the AR3 LA state, compared with the DA state, would appear to

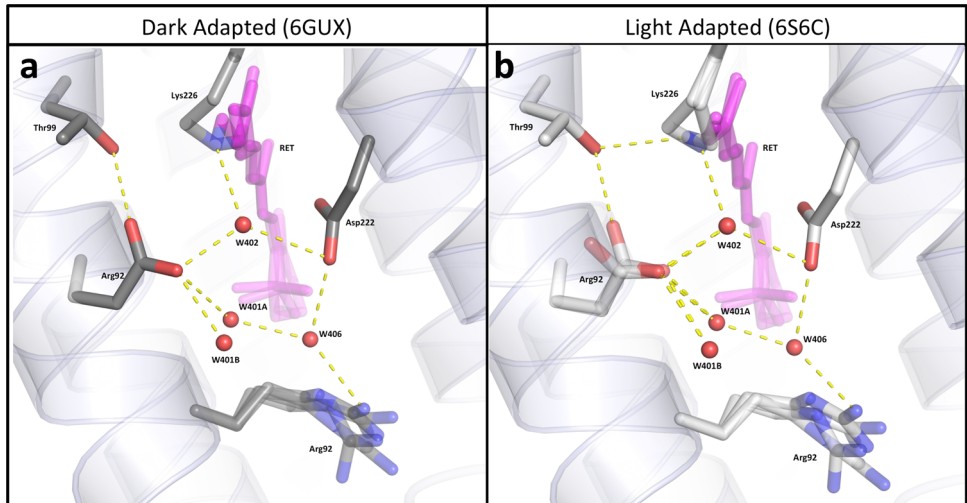

**Fig. 4 Structures of the pentagonal H-bond networks in AR3.** Predicted H-bonds are represented by yellow dashes (for distances see Supplementary Table 4) for DA (**a**) and LA AR3 (**b**). Selected amino acid sidechains are shown in stick representation with atoms colored using the CPK convention. Water molecules are shown as red spheres and retinal is colored pink. Wat401 is seen in two positions (A and B) at partial occupancy and the sidechain of Arg92 is seen in four conformations for both AR3 structures. Wat402 has single occupancy in AR3 (b) and bR (Wat602 in Supplementary Fig. 8b) in the LA state.

influence the order of the internal water molecules, both directly through W402 and indirectly via Thr99. Although W402 is well resolved in the 6GUX [https://doi.org/10.2210/pdb6GUX/pdb] structure, positive and negative electron density features are observed for this molecule in the 6S6C [https://doi.org/10.2210/pdb6S6C/pdb] structure confirming that it is more disordered in the LA state. In addition, in the AR3 LA state, the Asp95 side chain (which makes a direct H-bond to W402) is seen in two conformations and conversion between the two rotamers appears to involve the making/breaking of an H-bond with Thr99. It is possible that the stronger interaction that the SB N atom makes with the Thr99 side chain in the AR3 LA state, compensates for the partial breaking of the Asp95-Thr99 H-bond.

A second water molecule (W401) in the pentagonal network forms a H-bond to Asp95 and is also coupled to the chromophore isomerization state. In both AR3 structures, W401 is seen in two positions (A and B) ~1.5 Å apart (Fig. 4a, b), whereas in most bR ground-state structures[46], it is only present in one position (matching the W401B position in 6GUX [https://doi.org/10.2210/pdb6GUX/pdb] and 6S6C [https://doi.org/10.2210/pdb6S6C/pdb], Supplementary Fig. 9). In part, this difference in mobility appears to arise from the sequence Ser64-Ala65-Ala66 on Helix B of AR3, which replaces Phe67-Thr68-Met69 in bR and releases additional space to accommodate W401 oscillations. However, the mobility difference is also dependent on the order of the Asp95 side chain, since the relative occupancies of the two positions of W401 change between the two states of AR3. The Asp95-W401-W406 bond angle is increased to 146.5° in the LA state (compared to 140° in the DA state, Supplementary Table 5) further destabilizing the pentagonal H-bond network. The disorder observed in W401 (Supplementary Table 3) is consistent with previous Fourier-transform infrared spectroscopy (FTIR) experiments, which have suggested greater movement of W401 in AR3 than in bR[29]. It is interesting to note that W401 and Thr89$_{bR}$ are implicated in maintaining the Asp85$_{bR}$ sidechain pKa at ~2.2[47] and weaker equivalent bonds in AR3 would be consistent with a higher pKa for Asp95$_{AR3}$.

Classical MD simulations were used to evaluate the positions of water molecules in the vicinity of the SB, as observed in the crystal structures (Supplementary Figs. 9 and 10). An additional mobile water molecule is predicted in the vicinity of Asp95 in the

LA form and also possible in the DA-*cis* retinal structure. These discrepancies are interpreted as confirming the considerable disorder in the hydrogen-bonded water network, especially in the LA form, and could explain why larger currents are produced by the protein when expressed recombinantly in neurons, compared with other rhodopsins such as bR and halorhodopsin[13].

Although it does not directly form part of the pentagonal network of H-bonds, Arg92 stabilizes this arrangement by forming an H-bond to W406 from its Nε atom. In bR, the angle of the side chain (Arg82$_{bR}$) alters between the DA and LA states, pointing more towards the PRC in the latter (Supplementary Fig. 8)[46]. In contrast, in both the AR3 LA and DA states, Arg92 is resolved in four conformations, with similar occupancies and amplitudes of motion (Fig. 5). It therefore appears that changes in the rotamer populations of this residue do not provide a significant contribution to the transition between DA and LA states in AR3.

**Network of H-bonded water molecules between Arg92 and the PRC.** Several water molecules form a complex H-bond network between the two Glu residues (Glu204, Glu214) that together form the PRC on the extracellular face of AR3, and Arg92 (Fig. 5 and Supplementary Fig. 11). Several water molecules are resolved with partial occupancy, and the different conformations of Arg92 appear to stabilize different H-bonded networks that lead to the two Glu residues (Supplementary Fig. 11). The complexity of H-bonds in this region is possible because of a larger cavity in AR3 than bR. We suggest that this extensive water molecule network forms multiple interconnected pathways, which would appear to allow H⁺ transfer from the SB region to the PRC, independent of the orientation of the Arg92 sidechain. In bR, H⁺ release to the extracellular medium occurs during the M-state (which is itself divided into three substates)[48] and previous FTIR data have suggested that the differences in the organization of water molecules between bR and AR3 are even more pronounced in the M and N states than in the ground state[49]. We therefore suggest that the water-mediated parallel pathways for ion transfer, observed in the ground state, play a significant role in later photocycle stages and might reduce the time taken for H⁺ release and uptake.

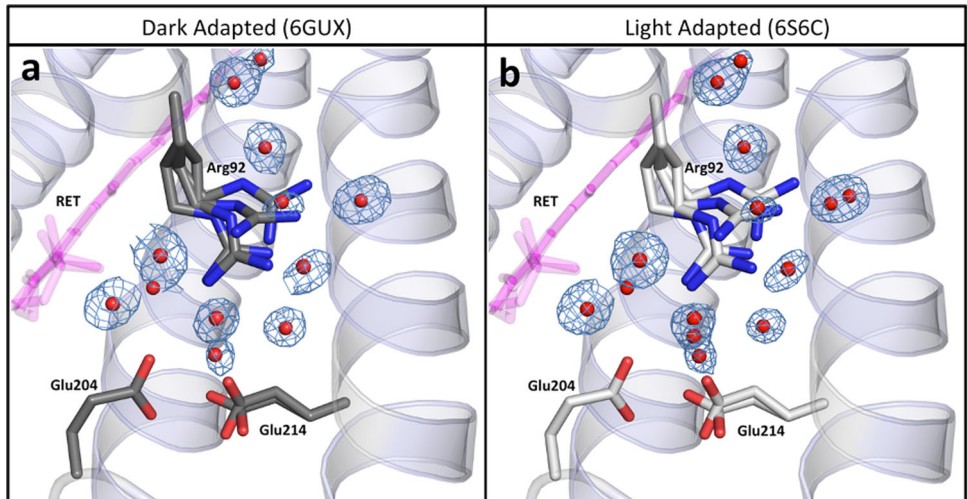

**Fig. 5 Structures of the Proton Release Complex (PRC) of AR3.** Glu204, Glu214 and the associated network of H-bonded water molecules in DA (6GUX [http://doi.org/10.2210/pdb6GUX/pdb]) (**a**) and LA AR3 (6S6C [http://doi.org/10.2210/pdb6S6C/pdb]) (**b**). Selected amino acid sidechains are shown in stick representation with atoms colored using the CPK convention. Water molecules are shown as red spheres and retinal is colored pink. The $2F_{obs}-F_{calc}$ electron density maps (blue mesh) around the water molecules are contoured at $1.2\sigma$.

The PRC is highly conserved across microbial rhodopsins. In a recent high-resolution structure of bR (5ZIM [https://doi.org/10.2210/pdb5ZIM/pdb]), both glutamate side chains are observed in different conformations, with rotations about the Cβ-Cγ and Cγ-Cδ bonds. In both the AR3 DA and LA structures, only Glu214 is present in two conformations (at 0.5 partial occupancies), distinguished by an 87.4° rotation around the Cγ-Cδ bond (Fig. 5). In AR3, the distances between the two Glu214 sidechain conformations and Glu204 is 2.3 Å (consistent with the distance previously reported for AR2) and 2.9 Å. Free-energy QM/MM calculations suggest that in both the DA and LA states, Glu214 is most likely to be protonated and that the Glu204 side chain is fully ionized, as observed in the ground state and early photo-intermediates in bR (Supplementary Fig. 12).

**Post-translational modifications and lipid binding.** Post-translational modifications of the protein were identified using nano electrospray ionization mass spectrometry (nESI MS) with protein solubilized in detergent (*n*-Octyl-β-glucoside) (Supplementary Fig. 13). An exact mass of 27,238 ± 0.9 Da for AR3 was determined, implying an extensively modified N terminus, with the removal of Met1-Leu6 and the conversion of Gln7 to a pyroglutamate (PCA) residue, as previously reported for bR[50,51] (Supplementary Figs. 13–15). Our AR3 structures reveal the post-translational modification of the N terminus, in agreement with the MS data. In addition, the nESI MS data suggests that the C terminus is also truncated with the removal of Asp258 (Supplementary Figs. 13 and 15); however, this region is not resolved in the structures presented here.

A non-covalent ligand with a mass of 1.06 kDa was observed bound to AR3 by MS and is most likely the archaeal lipid 1-*O*-[6''-sulfo-α-D-Mannosyl-1''-2'-α-D-Glucosyl]-*sn*-2,3-di-*O*-phytanylglycerol (S-DGD) (Supplementary Fig. 13b, c). This lipid is thought to stabilize the lipid bilayer during changes in osmolarity[52,53] and has been suggested as a proton donor for microbial rhodopsins[54,55]. S-DGD could not be resolved from the electron density of either of the AR3 structures presented here.

**Omega loop.** A so-called "omega loop structure" (formed by residues Asp11–Arg17) is observed in both crystal structures, the sequence of which (DLLxDGR) is highly conserved between archaerhodopsins (Fig. 6). This loop appears to form a binding

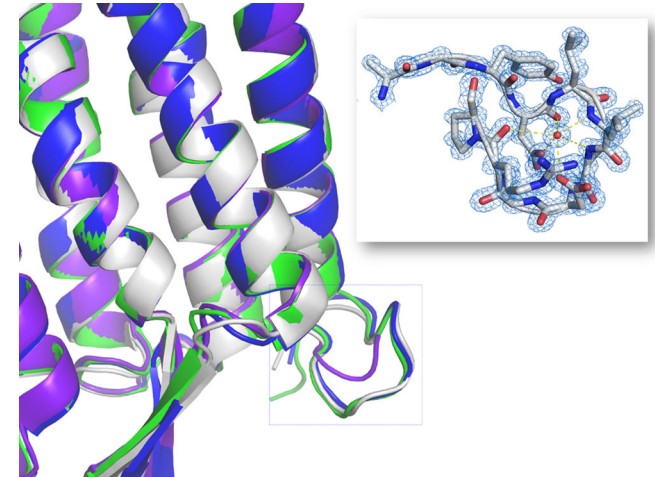

**Fig. 6 Comparison of the structures the N termini of AR3 and related microbial rhodopsins.** Overlay of structures of bR (5ZIM [http://doi.org/10.2210/pdb5ZIM/pdb] purple), AR1 (1UAZ [http://doi.org/10.2210/pdb1UAZ/pdb] green), AR2 (3WQJ [http://doi.org/10.2210/pdb3WQJ/pdb] blue), and LA AR3 (6S6C [http://doi.org/10.2210/pdb6S6C/pdb], white) showing the extracellular-facing omega loop, which is present in AR1, AR2, and AR3 but absent in bR. The inset shows details of the 6S6C [http://doi.org/10.2210/pdb6S6C/pdb] AR3 omega loop. Amino acids are shown in stick representation with atoms colored using the CPK convention. The $2F_{obs}-F_{calc}$ electron density map (blue mesh) is contoured at $2.3\sigma$.

site for an Na$^+$ ion, which coordinates with polar groups on the peptide backbone. During a 300 ns MD simulation, the ion was observed to exchange with water several times, without perturbing the conformation of the protein. We therefore conclude that Na$^+$ is not essential for maintenance of the loop structure, consistent with the deposited crystal structure of AR2 (3WQJ [https://doi.org/10.2210/pdb3WQJ/pdb]), in which water is modeled at this position[56].

In AR2, the omega loop has been proposed to form the intracellular end of the bacterioruberin binding site, which runs orthogonal to the membrane between helices E and F[32,33]. In the LA AR3 structure, the analogous site is partially occupied by lipid

tails and is occluded near the center of the membrane by the Phe150 side chain. We cannot exclude the possibility that the absence of the second chromophore in the structures reported here is an artifact of the crystallization process. Loss of bacterioruberin may also have induced monomerization of the protein during crystallization, although the crystals obtained do have the characteristic scarlet color observed in the native membranes (Supplementary Fig. 1).

## Discussion

The archaeal photoreceptor AR3 harvests energy from sunlight to transport H$^+$ ions from the cytoplasm of *H. sodomense* cells to the extracellular medium, creating a TM proton gradient for ATP synthesis. Like other members of the microbial rhodopsin superfamily, it has seven TM helices, which are arranged to create an internal channel linking the two sides of the membrane[37,38]. The crystal structures presented here, show that this channel is occluded by the retinylidene chromophore, which is formed by the post-translational, covalent conjugation of retinal to Lys226 via a SB. The water molecules within the channel form strong H-bonds to the conserved residues (including Arg92, Asp95, Asp106, Glu204, and Glu214), which are implicated in the mechanism of ion transport. Two AR3 structures have been determined. The first (6S6C [https://doi.org/10.2210/pdb6S6C/pdb]), which contains two forms of all-*trans* retinal, corresponds to the LA ground state. The second (6GUX [https://doi.org/10.2210/pdb6GUX/pdb]), which corresponds to the desensitized DA state, includes both 13-*cis* and all-*trans* retinal in a 7:3 isomer ratio.

A comprehensive understanding of the mechanism of photoreceptor desensitization has been hampered by the absence of high-resolution crystal structures of the DA forms of microbial rhodopsins. Long-standing questions in the field have concerned the energetics of the transformation from the LA ground state to the DA state and, in particular, how the thermodynamic equilibrium between all-*trans* and 13-*cis* retinal in the DA state is established in the absence of light and with no apparent input of energy. (There is limited structural data for the DA states of other microbial rhodopsins and the only DA crystal structure currently deposited in the PDB is for bR[27,57].) It is perhaps surprising that, in AR3, there are minimal changes in the positions of the side chains that line the retinal binding pocket between the two states and it would therefore appear that the favoring of the *cis* isomer in the DA state, does not arise from changes in the binding energy of the chromophore. Instead, our QM/MM calculations (which use the crystal structures as a starting point) show that the subtle changes in partial charge distribution around the SB (in the low dielectric environment of the center of the membrane) and the lower mobility of the Lys226 sidechain in the DA form, reduce the activation energy for retinal isomerization (compared to the LA form). Our calculations also show, that the difference in energy between *cis* and *trans* retinal is approximately ten times less in the DA form than the LA form, consistent with the thermodynamic equilibrium between the two isomers, observed in the DA states of several microbial rhodopsins.

These crystal structures also allow us to gain a better understanding of the extent to which the conformation of the chromophore is coupled to the networks of internal water molecules. The movement of the SB nitrogen atom in the LA state appears to destabilize the quasi-planar pentagonal network of H-bonded groups—directly via W402 and indirectly via Thr99 and Asp95. From this study alone, it is not clear the extent to which the observed differences in this region between the DA and LA states are important for the mechanism of proton pumping. Although we can be confident from the all-*trans* retinal, that the LA crystal

structure presented here does indeed correspond to the AR3 ground state, it shares some features with the early intermediates of the bR photocycle: first, time-resolved X-ray free-electron laser experiments on bR have shown that, in the 100 s fs after light absorption, Wat402 is displaced away from the SB nitrogen atom[43]. Second, the H-bond between Thr89$_{bR}$ and Asp85$_{bR}$ (Thr99$_{AR3}$-Asp95$_{AR3}$), which breaks during the formation of the bR M-state[41], is already partially broken in the ground state LA AR3 structure. Third, Wat401 becomes disordered within 13.8 µs of photon absorption in bR[42], whereas this molecule is resolved in two distinct positions in both AR3 structures reported here. We would therefore suggest that the AR3 ground state is "more advanced" in photocycle terms than the bR ground state and that this might have an influence on the kinetics of ion transport.

Atomic force microscopy images of the *H. sodomense* claret membrane (Supplementary Fig. 4) indicate that AR3 is a trimer in vivo; however, the structures presented here are monomeric and were obtained from a three-dimensional crystal in which packing between adjacent molecules differs from that of the native environment. Several studies have examined the functional differences between monomeric and trimeric forms of the microbial rhodopsins and of bR in particular. The dissociation of oligomers is generally characterized by a hypsochromic shift in the absorption spectrum, which is primarily caused by the loss of exciton coupling between the chromophores, rather than by structural changes within the monomers[58–60]. Although the photocycles of monomeric and trimeric bR are qualitatively the same[61,62], the kinetics of individual steps vary, in particular those later stages that involve larger conformational changes[63]. Dencher et al.[64] reported that the equilibrium position between 13-*cis* and all-*trans* retinal in DA bR is perturbed by monomerization. The dissociation of AR3 oligomers will inevitably alter the interactions between the individual molecules and their environment. This may, in turn, perturb the behavior of dynamic structural elements within the interior of the protein, including amino acid side chains (thereby influencing their protonation state), water molecules and the chromophore itself. It is therefore essential that structures of DA microbial rhodopsins, crystallized as trimers, be obtained in order to determine the influence of oligomerization state on the mechanisms of receptor desensitization and resensitzation.

We have attempted to mitigate the effects that the crystallization process might have on the LA and DA states of AR3, by preparing wild-type protein, expressed in its native organism and purified without any detergents that might remove closely associated native lipids and bacterioruberin. We have crystallized the protein in LCP, to provide an environment which more closely resembles the native membrane than can be provided by micelle-based methods. Although our QM/MM calculations are in qualitative agreement with the expected chromophore isomer ratio, we cannot exclude the possibility that crystal contacts influence the conformational flexibility of the protein and that this might give rise to non-physiological behavior.

The structures of DA and LA AR3 presented here, also have implications for our wider understanding of the process of desensitization in other receptor families. They highlight how minimal displacements of charged and hydrophilic groups within the low dielectric environment of the membrane can induce changes in ligand conformation and vice versa. Finally, these structures also provide information that increases our understanding of the mechanism of H$^+$ translocation by AR3, and will facilitate the design of further, more efficient Arch mutants for applications in optogenetics.

## Methods

**Protein expression and purification.** *H. sodomense* (ATCC-33755) cells were purchased from LGC Standards Ltd (Teddington, UK) and were grown without any genetic modification (see Supplementary Information). Cells were collected by centrifugation (8000 × g; 30 min; 4 °C) and the pellets were resuspended in 4 M NaCl and DNAse I (Sigma, UK). The solution was stirred for 2 h before being manually homogenized. The preparation was dialyzed overnight in 0.1 M NaCl, and centrifuged (70,000 × g; 50 min; 4 °C). Sucrose density gradient ultra-centrifugation was used to isolate the AR3-rich membrane, using a step gradient consisting of layer of 4 mL of sucrose at densities of 30, 40, 50, and 60% w/v, and centrifuged at 110,000 × g for 15 h at 15 °C. The lower band with a pink/purple color was collected, and the sucrose remaining in the sample was removed through overnight dialysis against distilled water. The sample was then further centrifuged (70,000 × g, 50 min, 4 °C), and the pellet was resuspended in distilled water to a final concentration of 20 mg/ml. Using SDS-polyacrylamide gel electrophoresis, the AR3 content of the samples was estimated to be 78 ± 2% (w/w) of the total protein, which is comparable to bR purified by the same method[65,66]. Samples were stored at 4 °C prior to spectroscopic experiments and crystallization.

**Crystallization in LCP.** The non-delipidated AR3 protein sample was mixed with molten monoolein lipid (Nu-Check) in a 40:60 volume ratio using two gas-tight Hamilton syringes connected by a TTP Labtech syringe coupler[67]. The LCP mixture was dispensed onto a 96-well glass crystallization plate using a TTP LCP-Mosquito crystallization robot (TTP Labtech) at 18/1 ratio (540 nL reservoir + 30 nL of LCP). The LCP plate was sealed with a glass cover and stored at 20 °C. All crystallization procedures were performed under dim light.

Crystals of AR3 appeared after 2–3 days in a precipitant solution containing 30% v/v polyethylene glycol 600 (Fluka Analytical), 100 mM MES buffer pH 5.5, 150 mM NaCl, and 150 mM Ca²⁺ (Supplementary Fig. 1a). Crystals were collected using Dual-Thickness MicroMounts *cryo*-loops (MiTeGen, Ithaca USA), then flash-frozen and stored in liquid nitrogen.

**X-ray data collection and processing.** X-ray data were collected from different crystals at 100 K on the I24 microfocus beamline at Diamond Light Source (Harwell, UK) using a beam size of 6 μm × 9 μm and a Pilatus3 6 M detector (DECTRIS).

Diffraction patterns were integrated using DIALS (version 1.10.1) and several data sets were all combined using BLEND (version 0.6.23)[68] with default parameters. The diffraction data from individual crystals were integrated using the Xia2 pipeline (0.3.8.0)[69] running DIALS and then, in a separate step, merged using AIMLESS (version 0.0.14)[70]. Phases for the AR3 *cryo* structures were obtained by molecular replacement using the Phaser software (version 2.7.17) from the CCP4 Suite (version 7.0.066)[71] with the deposited structure for AR1 (1UAZ [https://doi.org/10.2210/pdb1UAZ/pdb][32]) as the search model. The initial electron density maps were inspected, and the model was built using Coot (0.8.9.1)[72,73]. The structure models were refined using PHENIX (version 1.18.2)[74] and Refmac5[75]. Retinal occupancy ratio for both structures were determined using several tools such as occupancy refinement in PHENIX[74], observation of the retinal B factors before and after refinement (using different occupancies values), observation of the calculated m$F_{obs}$–$DF_{calc}$ maps corresponding to each occupancy value and ligand validation tools in PHENIX, Coot, and wwPDB[76,77]. In the deposited PDB files, amino acid residues were numbered according to the deposited UNIPROT sequence BACR3_HALSD and water molecules were numbered following the convention used in the 1C3W [https://doi.org/10.2210/pdb1C3W/pdb] structure for bR.

**QM/MM optimization.** Each of the two retinal conformations found in the DA and LA crystal structures were optimized with a hybrid QM/MM approach using the ChemShell software package (version 3.7.0)[78,79]. The QM region contained retinal and the side chain of Lys216, while the MM region consisted of the remainder of the protein and crystal waters. The B3LYP[79–83] functional with the cc-pVDZ[84,85] basis set was used as the QM method. The MM region was described with the CHARMM36 protein force field[86] and the TIP3P water model[87]. During the optimization only the QM region was allowed to relax, while all other atoms were constrained to their crystal positions.

**Computation of potentials of mean force.** PMFs were generated by using QM/MM in combination with the Weighted Histogram Approach Method (WHAM)[88] as implemented by Alan Grossfield[89]. This approach was used in earlier work that utilized WHAM for proton transfer computations[90] and a more detailed description of the methodology can be found there. To ensure sufficient sampling of the reaction coordinates, the self-consistent charge density functional tight binding method with full third-order extension[91–95] was applied to the QM region. The MM region used the CHARMM protein force field[86] and TIP3P water model[87].

For the proton transfer between Glu204 and Glu214, the reaction coordinate was defined as the difference between donor–hydrogen distance and acceptor–hydrogen distance. For the retinal isomerization the reaction coordinate was defined as the C12–C13–C14–C15 dihedral of retinal. The reaction pathways

were sampled for a reaction coordinate of −1.7 to 1.7 Å and from −90° to 270° for proton transfer and retinal isomerization, respectively. The reaction coordinate was restrained with a harmonic potential of 150 kcal/mol and sampled in steps of 0.1 Å for proton transfer and 5° for retinal isomerization, with each point being sampled for 0.5 ns. The final PMF profiles were obtained by combing sampling initiated from the 13-*cis* and all-*trans* retinal conformation in the case of 6GUX [https://doi.org/10.2210/pdb6GUX/pdb] and from the minor and major *trans* states in the case of 6S6C [https://doi.org/10.2210/pdb6S6C/pdb].

**Reporting summary.** Further information on research design is available in the Nature Research Reporting Summary linked to this article.

## Data availability

Structures and diffraction data have been deposited in the protein databank (https://www.rcsb.org). The accession codes are 6GUX [https://doi.org/10.2210/pdb6GUX/pdb] for dark-adapted AR3 and 6S6C [https://doi.org/10.2210/pdb6S6C/pdb] for light-adapted AR3. Mass spectrometry data are available for download [https://doi.org/10.6084/m9.figshare.13293203.v1]. Other data are available from the corresponding authors upon reasonable request.

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

## Acknowledgements
We thank Dr. Robin Owen and Dr. Darren Sherrell, and the I24 beamline staff (Diamond Light Source) for their support during data collection under the MX proposals 19152 and 11386. We thank Juan Escobar and Peter Fisher (Oxford) for technical assistance, and Dr. Agata Butryn (Diamond Light Source) and Dr Rosana Reis (NPL) for helpful discussions. We are grateful for support from the Membrane Protein Laboratory under Wellcome Trust grant number 20289/Z16/Z, including the award of experimental time (SM15222) on the B23 Beamline at Diamond Light Source (UK) and we acknowledge the support of Dr. Giuliano Siligardi, Dr. Rohanah Hussain and Dr. Charlotte Hughes. We acknowledge funding from United Kingdom Department of Business, Energy and Industrial Strategy (BEIS) to I.M., from the European Research Council (ERC) under the European Union's Horizon 2020 research and innovation program (Grant Agreement number 678169, ERC Starting Grant "PhotoMutant") to I.S., from DSTL UK (grant number DSTLX-1000099768) and BBSRC (grant number BB/N006011/1) to A.W. We thank the Minerva Stiftung for a post-doctoral fellowship within the framework of the Minerva Fellowship Program to SA and the DFG for a Mercator Fellowship to I.S. (Grant number SFB 1078).

## Author contributions
P.J.J., I.M., and A.W. conceived the study and all authors were involved in designing experiments and interpreting data. AR3 protein was expressed and purified by J.F.B.J. and J.V. Crystallization was performed by J.F.B.J., P.J.J., J.B., T.O.C.K., and I.M. Diffraction data were acquired by J.F.B.J., P.J.J., D.A., and I.M. High-resolution structures were solved by P.J.J., D.A., and I.M. Simulations were performed by S.A. and I.S. Mass spectrometry data were acquired by K.K.H., H.Y.Y., and C.V.R. Atomic force microscopy images were acquired by A.V. and P.E.M. J.F.B.J., P.J.J., I.M., and A.W. wrote the paper with assistance from all authors. J.F.B.J. and P.J.J. contributed equally to this paper.

## Competing interests
The authors declare no competing interests.
