## [Peer Review File · Nature Communications]

REVIEWER COMMENTS

Reviewer #1 (Remarks to the Author):

The authors report two new high-resolution x-ray structures for the archeorhodopsin-3 proton transporter. The major findings are that, in the desensitized, dark-adapted state versus the sensitized, light-adapted state, there are only minimal changes in the side-chain positions. In the dark-adapted state, both 13-cis and all-trans retinal isomers are found in a 3:1 isomer ratio, which is well known from studies of other microbial rhodopsins. In the light-adapted state, only the all-trans retinal isomer is present, albeit with two different conformations for the positions of the Lysine226 side chain and the Schiff base nitrogen atom.

Based on QM-MM simulations, the favoring of the cis-isomer in the dark-adapted state does not entail the binding energy of the chromophore. Rather, changes in the partial charge distribution about the Schiff base are involved, leading to an ~tenfold difference between the cis- and trans-retinal isomers in the dark-adapted state. In the dark-adapted state the retinal isomers achieve their equilibrium distribution, while the protein environment in the light-adapted state selectively favors the trans isomer. All of these results are significant to the field.

Perhaps most significant, the conformation of the retinal chromophore is coupled to networks of internal water molecules. The light-adapted state destabilizes or disorders a pentagonal network of hydrogen-bonded groups, involving three key water molecules. Comparison to bacteriorhodopsin suggest that the kinetics of proton transport are associated with disordering of the internal water networks. They include the pentagonal hydrogen-bonded network about the beta-ionone ring, as well as the network of hydrogen-bonded water molecules in the proton release complex in the vicinity of the retinal. Disordering of the water molecules in the light-adapted state is suggested to be correlated with an advancement along the photocycle reaction coordinate versus other microbial rhodopsins, leading to faster photocycle kinetics for archeorhodopsin-3.

Major points:

1. The results are novel and the manuscript is important to scientists in field of retinal proteins and biomembranes. The claims entail careful x-ray crystal analysis, supplemented by native mass spectrometry, QM-MM simulations, and spectral characterization the samples, and are convincing. All of the work is carefully performed by leading investigators, with careful data analysis and reduction. The findings are appropriately discussed in the context of the previous literature, and the paper will contribute to thinking in the field.
2. The manuscript is clearly written and is sufficiently concise to communicate the most important findings. The authors present the strengths of their findings, and although the comments about broad implications for desensitization of other receptors is a bit of a stretch, it is not beyond the reach of their conclusions. The treatment of the previous literature in the field is balanced and fair, and the authors provide ample methodological detail, such that the experiments can be reproduced by other laboratories. The statistical analysis of the data is sound, and the protein structures have been made publicly available in the Protein Data Bank.
3. The strength of the paper is that very high-resolution structures have been determined, including the desensitized state of the photoreceptor, which are corroborated by native mass spectrometry, atomic force microscopy, and QM-MM simulations. The careful and extensive Supplementary Information is a plus.
4. The weakness is that many of the results are largely confirmatory of what is already well- known for other microbial rhodopsins, which have likewise been extensively studied. These features include

the multiple retinal conformers in the dark-adapted, desensitized state, as well as the pentagonal hydrogen-bonded network, and the hydrogen-bonded network and associated water molecules about the proton release complex. The implications for receptor desensitization are emphasized as a potential novel aspect of the findings. Regardless, the contribution to the corpus of structural knowledge of microbial rhodopsins and the careful data reduction and analysis are viewed to be significant to the field.

5. Supplementary information, Figure 3 - the far-UV CD spectrum does not have any noise, even at low wavelengths. Please clarify whether it is an experimental spectrum.

Minor points:

1. Page 1, abstract - please clarify that what is meant by the ground state is the light-adapted state.
 2. Page 3, bottom paragraph - the first sentence reads as if AR3 is a GPCR which it is not.
 3. Page 9, second paragraph - please clarify that what is actually observed is the distribution of water molecules, which is static in x-ray crystallography, not the dynamics.
 4. Page 14, second paragraph - please clarify for which other cases of microbial rhodopsins a dark-adapted crystal structure is available.
 5. Supplemental information, Page 10, Figure 9 - the authors may wish to make reference to molecular dynamics simulations of other photoreceptors, which analogously probe changes in water density due to retinal light absorption, for example the work of Grossfield (Leioatts et al. (2014), Retinal ligand mobility explains internal hydration and reconciles active rhodopsin structures, *Biochemistry* 53, 376–385).
- Michael F. Brown

Reviewer #2 (Remarks to the Author):

In this paper, the authors report the crystal structures of the dark-adapted (DA) and light-adapted (LA) states of the wild-type Archaerhodopsin-3 (AR3) photoreceptor, solved to 1.3 Å and 1.1 Å respectively. They observed differences in the chromophore structures and those in the positions of the internal water molecules between the two states. The crystal data are well analyzed. However, insights derived from the structural data are very few in this paper. The manuscript does not contain clear descriptions of how the structures and structural differences in the DA and LA states are related to understanding the functional mechanism of AR3. Accordingly, this work is premature to be published. I do not recommend the publication of this work in Nature Communications.

Reviewer #3 (Remarks to the Author):

The authors report in this manuscript on a phenomenal achievement: structural features of a microbial rhodopsin in the dark-adapted and light-adapted state at an unprecedented resolution of 1.3 and 1.1 Å, respectively. This brings a wealth of novel molecular details, that are interpreted with help of an impressive array of biophysical techniques and in-silico simulations. All-in-all a groundbreaking manuscript. Interesting novel data are also presented with respect to posttranslational modifications of Archaerhodopsin-3, but the most essential message of course comprises the structural alterations recognized within and between the DA and LA states. However, a major concern is that all data have been obtained under conditions (monomeric state in a very restrictive crystalline environment) that

are far off from the trimeric state in the native claret membrane. The authors care-free translate their observations into a native perspective. In my view it is essential that the authors add a section clearly discussing and as much as possible substantiating to what extent their data and interpretation can be translated to the native membrane environment.

Major comments:

1. With an about 2:1 occupancy of the 13-cis and all-trans chromophore in the DA state (page 6, lines 4-6) I would expect a free energy difference for the all-trans <-> 13-cis equilibrium of about - 0.4 kcal/mol. This is much smaller than the calculated free energy difference (- 1.9 kcal/mol; page 8, top). Hence, the presented energy differences should only be considered as quite rough indicators for the all-trans/13-cis equilibrium.

2. It is suggested (page 8, lines 7-9), that in the LA state thermal all-trans to 13-cis isomerization does not occur because of the calculated higher activation energy compared to the DA state. However, this is not correct in my opinion. The free energy difference calculated for the all-trans/13-cis equilibrium in the LA state of about 10.9 kcal/mol is so high, that even if the real energy difference would be only half this value, the 13-cis population in this equilibrium would be negligible, irrespective of the height of the activation barrier.

3. Has it been taken into account that the thermal all-trans to 13-cis isomerization in the DA state comprises a reaction differing from the light-dependent one in the LA state ? The first one is a 13-trans,15-anti to 13-cis,15-syn transition (the SB proton should remain in roughly the same position), while the latter is a 13-trans,15-anti to 13-cis,15-anti transition (the SB proton can then be transferred to the proton-accepting residue).

4. The authors imply (page 7, lines 4-7 from bottom) that upon light activation of the DA state only the all-trans population enters the photocycle with active proton pumping, while the depleting all-trans population is then repopulated via the thermal equilibrium with the 13-cis state. Considering the quite high activation barrier calculated for this equilibrium (17 kcal/mol; page 8, line 7), the rate of this thermal repopulation is relatively low. It is questionable whether in this way within two minutes the 13-cis state can be fully converted into all-trans, as can be clearly concluded from analysis of the LA crystals. To me a more realistic view is that the 13-cis population (13-cis,15-syn chromophore) is also excited by illumination, generating a 13-trans,15-syn state, which thermally rapidly relaxes to the 13-trans,15-anti state.

5. Do the calculated free energy differences (page 8, top half) and C13=C14 dihedral-dependent energy levels (Fig. 3) pertain to the major or the minor trans state in the LA case ? Are there significant differences in these aspects between these two trans states ?

6. A major concern is that apparently during crystallization the native trimer organization is broken up into monomers that generate the crystal unit cell. For several microbial rhodopsins it has been reported, that the pump function may not be severely affected by monomerization, but biochemical and biophysical properties like the thermal stability and absorbance maximum are affected (e.g. Brouillette et al. (1989) *Proteins* 5(1): 38-46; Ranaghan et al. (2011) *JAmChemSoc* 133(45): 18318-18327; Tsukamoto et al. (2014) *JPhysChemB* 118(43): 12383-12394; Hussain et al. (2015) *JMolBiol* 427(6): 1278-1290; Kao et al. (2019) *JPhysChemB* 123(9): 2032-2039; Idso et al. (2019) *JPhysChemB* 123(19): 4180-4192; Ganapathy et al. (2020) *BBA* 1862: 183113). This implies also subtle structural rearrangements in the protein including the chromophore. Hence, it is conceivable that the water structure and distribution in the protein and around the SB region is perturbed upon monomerization. This aspect needs proper discussion, and the authors should present evidence that the structural differences observed between the all-trans and 13-cis DA states are not, or at the most only partially due to monomerization.

7. A similar concern pertains to the LA state, but there an additional caveat arises in the non-native environment in which the LA state was generated. The protein-protein and protein-lipid contacts in a monomeric crystal of course differ dramatically from the native trimeric organization in the claret membrane and present quite unnatural boundary conditions and dynamics. Hence, I consider it essential that the authors also for the LA state discuss to what extent this can obstruct the translation of their interpretations to the native trimeric organization. For instance, can the restricted dynamics in the crystalline state, together with the excess heat produced during excitation, which cannot be properly absorbed by the protein, result in artifactual generation of two all-trans states ?

Minor comments:

8. The authors claim that AR3 generates a much faster photocycle and larger current than bR (page 3, line 8 from bottom; pages 10 and 11, bottom lines). However, there is contradictory evidence about the kinetics and pump activity of AR3 versus bR. For instance, Vogt et al. record about the same photocurrent for AR3 and bR (Vogt et al. (2013) *BiophysJ* 105(9): 2055-2063).

9. Re: chromophore configuration in photocycle intermediates (page 4, lines 5-6): most photo-intermediates do not contain the 13-cis,15-syn configuration, but either the 13-cis,15-anti (early phase) or the 13-trans,15-anti (late phase) configuration (see comment 3 and Naito et al. (2019) *BiophysRev* 11(2): 167-181).

10. AR3 was purified by isolating claret membrane vesicles. The Methods section should present the AR3 purity level in this preparation in % w/w of the protein content.

11. The purified AR3 absorbance spectrum (Fig. S3b, black curve) has a strange asymmetrical shape. Has it been baseline overcorrected, or does it still contain some bacterioruberin ?

12. Textual comments:

a. Not all microbial rhodopsins undergo dark-adaptation. Replace "are" (Abstract, line 2) by e.g. "contain".

b. Replace "Although the wild-type protein is more usually classified as a light-driven transport protein, " (page 3, lines 16-17) by "Although the wild-type protein is classified as a light-driven proton pump, ".

c. Insert "or as membrane voltage sensors" after "specific wavelengths" (page 3, line 9 from bottom). This application has become much more important than altering neural activity.

d. All through the manuscript the chromophore of AR3 is named "retinal". Indeed, the chromophore is generated by covalently binding retinal to a lysine amino group via a Schiff base, but the resulting entity is a derivative of retinal and is called a retinylidene moiety. Better replace in all these cases "retinal" by "chromophore" or "retinylidene element".

e. As stated at page 6, line 4, the modeled 13-cis to all-trans ratio in the DA state is 70 to 30%. Please correct the 3:1 ratio mentioned at page 14, line 3, accordingly.

f. The sentence "Repeated bindingas water." (page 15, lines 7-10) is not clear to me. Why would repeated binding have a larger effect on very rapidly reacting systems like H-bonded networks than a single binding event ? In my view, this sentence does not add much to the statement in the previous sentence. Hence, I would recommend to omit the sentence starting with "Repeated binding..." or else clarify it better.

g. References 24 and 44 are the same, except that the volume number in 24 is incorrect.

h. The two W402 residue numbers in the legend of Fig. S7 should be corrected to W602.

Reviewer #4 (Remarks to the Author):

Archaeorhodopsins are light-activated photoreceptors that are prominent representatives of microbial rhodopsins, which have recently gained particular interest in the field of optogenetics where they enable control over the stimulation or inhibition of individual neurons. Archaeorhodopsin-3 (AR3) catalyzes the light-driven transport of protons similar to its well-characterized homologue bacteriorhodopsin (BR). Juarez et al. have solved the crystal structures of AR3 in its ground (or light-adapted) state and in its dark-adapted state to outstanding resolutions of 1.1 Å and 1.3 Å. Based on these high-resolution structures the authors have analysed the differences in molecular interactions within and around the retinal binding pocket. Together with computational analyses their data provide detailed explanations for an observed thermodynamic equilibrium between the different chromophore conformations that is proposed as a molecular basis for photoreceptor desensitization.

The deposited crystal structures represent the first high-resolution structural data for AR3, giving detailed insight into the proposed proton translocation pathway. In addition, and probably more importantly, the analysis and comparison of the ground and desensitized dark-adapted state provide evidence for the molecular mechanism of photoreceptor desensitisation. These novel structural insights are valuable contributions toward a complete understanding of the mechanism of proton translocation in microbial rhodopsins and further facilitate the application of archaeorhodopsins in optogenetics.

Overall, the submitted study is presented in a clear fashion and the data were thoroughly analyzed, leading to reasonable and convincing conclusions. The novelty of the findings and their significance for the microbial rhodopsin community and the field of optogenetics warrant publication of this work after some revisions.

The following issues should be addressed:

- The authors mention the faster photocycle kinetics of AR3 compared to its homologs and relate this phenomenon to distinct H-bonded networks around the proton release complex (PRC) that could provide multiple pathways for proton transfer. This is an interesting observation and hypothesis. However, from the text it is not obvious what underlying molecular mechanism the authors hold responsible for this effect. Am I correct to assume that this proposition is based on the fact that proton uptake and release have been shown to be rate-limiting steps in the photocycle of microbial rhodopsins and that multiple exits could reduce this bottle-neck effect? I would suggest to elaborate this point a bit more.

- On the same topic, a quick literature search revealed a study from Geng et al. (Photochemistry and photobiology, 2018) that discussed the importance of the highly conserved residues T164 and S165 (nomenclature in HeAR) in the faster turnover of ARs, which was not discussed in the presented manuscript. Do the novel AR3 structures provide any structural evidence that could corroborate this observation?

- As discussed, a similar distribution of all-trans and 13-cis retinal in the dark-adapted state is observed in BR. For the latter, isomeric composition has been found to be dependent, among other important factors, on the oligomeric state of the protein. Significantly altered rates for light-dark adaptation were observed for monomeric BR compared to the oligomer indicating certain regulatory interactions between neighboring protomers (Dencher et al., Biochemistry, 1983). Cross-protomer interactions that can directly affect the photocycle have also been demonstrated in the crystal

structure of the blue-light absorbing proteorhodopsin (Ran et al., Acta Crystallographica Section D, 2013). Since the AR3 crystal structures was solved in the monomeric state such potential interactions that could occur in the native membrane environment cannot be observed. It would be interesting to investigate this possibility by comparing or combining the structural data of the dark- and light-adapted state from AR3 with the information from the trimeric structure of AR2 for example.

- The omega loop in AR3 seems to coordinate a sodium ion. Is this coordination required for the stabilization of the specific loop conformation, which in turn possibly enables binding of bacterioruberin? Interestingly, in the AR2 structure at 1.8Å (3WQJ) a water molecule was modeled in the same position.

- The AFM topograph in Supplementary Figure 4 is of low quality and misleading (probably recorded with an asymmetric tip). Please provide a topograph of higher quality and only display a region of the image with lattice lines (currently, the structure of the topograph is not easily discernible).

- For Supplementary Table 2, please use values (a, b, alpha) from images recorded from different adsorption of 2D crystals on mica. It could be that the used piezo scanner has a creep (i.e., in slow scan direction) – please check if AFM scanner is well calibrated (please indicate used scanner in Materials and Methods). Alternatively, the authors might record images of negatively stained 2D crystals by electron microscopy, and calculate and analyse power spectra to prove that vectors a and b are indeed different. Finally, please check numbers, e.g., of a and b (vectors) in Supplementary Table 2 and legend to Supplementary Figure 4 – the rounding is not consistent.

- The abbreviation AFM is not used in manuscript, please remove from “Abbreviation List”.

Response to Reviewers' Comments

We thank the reviewers for their comments, recommendations and suggestions, and are pleased that there are acknowledgements of:

- the exceptionally high resolution (one is higher than for any other membrane protein reported to date) of these structures, as recognized by the majority of reviewers;
- one structure being only the second "dark-adapted" structure for a photoreceptor to be published, giving new insights into the energetics of retinal isomerization;
- these being the first crystal structures ever of AR3 (monomeric or trimeric), after many attempts worldwide, with AR3 being an essential component of so much optogenetics work thereby raising the level and significance of the work.

In addition to the points noted below, we have corrected some minor typographical errors.

Reviewer #1:

1. The results are novel and the manuscript is important to scientists in field of retinal proteins and biomembranes. The claims entail careful x-ray crystal analysis, supplemented by native mass spectrometry, QM-MM simulations, and spectral characterization the samples, and are convincing. All of the work is carefully performed by leading investigators, with careful data analysis and reduction. The findings are appropriately discussed in the context of the previous literature, and the paper will contribute to thinking in the field.

Authors' response: We thank the reviewer for these comments.

2. The manuscript is clearly written and is sufficiently concise to communicate the most important findings. The authors present the strengths of their findings, and although the comments about broad implications for desensitization of other receptors is a bit of a stretch, it is not beyond the reach of their conclusions. The treatment of the previous literature in the field is balanced and fair, and the authors provide ample methodological detail, such that the experiments can be reproduced by other laboratories. The statistical analysis of the data is sound, and the protein structures have been made publicly available in the Protein Data Bank.

Authors' response: We thank the reviewer for these comments.

3. The strength of the paper is that very high-resolution structures have been determined, including the desensitized state of the photoreceptor, which are corroborated by native mass spectrometry, atomic force microscopy, and QM-MM simulations. The careful and extensive Supplementary Information is a plus.

Authors' response: We thank the reviewer for these comments.

4. The weakness is that many of the results are largely confirmatory of what is already well-known for other microbial rhodopsins, which have likewise been extensively studied. These features include the multiple retinal conformers in the dark-adapted, desensitized state, as well as the pentagonal hydrogen-bonded network, and the hydrogen-bonded network and associated water molecules about the proton release complex. The implications for receptor desensitization are emphasized as a potential novel aspect of the findings. Regardless, the contribution to the corpus of structural knowledge of microbial rhodopsins and the careful data reduction and analysis are viewed to be significant to the field.

Authors' response: We are grateful to the reviewer for recognising that our data and analysis are significant contributions to the field.

5. Supplementary information, Figure 3 - the far-UV CD spectrum does not have any noise, even at low wavelengths. Please clarify whether it is an experimental spectrum.

Authors' response: This is an experimental spectrum. At a late stage in the preparation of the manuscript, the CD spectrum collected on a benchtop spectrometer was switched for one acquired at a synchrotron source, however the methods section was not updated. We apologise for this error and are grateful to the reviewer for identifying this mistake. We have corrected the methods section and the figure caption to make it clear that this is an experimental spectrum.

Minor points:

1. Page 1, abstract - please clarify that what is meant by the ground state is the light-adapted state.

Authors' response: We have made a change to clarify this point.

2. Page 3, bottom paragraph - the first sentence reads as if AR3 is a GPCR which it is not.

Authors' response: We have modified the first sentence to make this clear.

3. Page 9, second paragraph - please clarify that what is actually observed is the distribution of water molecules, which is static in x-ray crystallography, not the dynamics.

Authors' response: We have added an additional sentence to make this clear.

4. Page 14, second paragraph - please clarify for which other cases of microbial rhodopsins a dark-adapted crystal structure is available.

We are only aware of the 1X05 bacteriorhodopsin structure and we have added a citation to the relevant paper at the appropriate point in the text - Nishikawa *et al.* (2005).

5. Supplemental information, Page 10, Figure 9 - the authors may wish to make reference to molecular dynamics simulations of other photoreceptors, which analogously probe changes in water density due to retinal light absorption, for example the work of Grossfield (Leioatts et al. (2014), Retinal ligand mobility explains internal hydration and reconciles active rhodopsin structures, *Biochemistry* 53, 376–385).

Authors' response: We have added a citation to this study on page 9 of the main paper and we thank the reviewer for drawing our attention to this work.

Reviewer #2:

In this paper, the authors report the crystal structures of the dark-adapted (DA) and light-adapted (LA) states of the wild-type Archaerhodopsin-3 (AR3) photoreceptor, solved to 1.3 Å and 1.1 Å respectively. They observed differences in the chromophore structures and those in the positions of the internal water molecules between the two states. The crystal data are well analyzed. However, insights derived from the structural data are very few in this paper. The manuscript does not contain clear descriptions of how the structures and structural differences in the DA and LA states are related to understanding the functional mechanism of AR3. Accordingly, this work is premature to be published. I do not recommend the publication of this work in *Nature Communications*.

Authors' response: We respectfully disagree with the suggestion that there are very few insights derived from the structural data. In particular we believe that these structures enable us to resolve significant and long-standing questions in the field as to the energetics of the transition from light-adapted to dark-adapted (DA) states in particular, how the thermodynamic equilibrium between all-*trans* and 13-*cis* retinal in the DA state is established in the absence of light and with no apparent input of energy. We would also note that a full understanding of the molecular mechanisms of H⁺ ion transport by AR3 would require structural characterisation of photocycle intermediates, which is beyond the scope of this present study.

Reviewer #3 (Remarks to the Author):

The authors report in this manuscript on a phenomenal achievement: structural features of a microbial rhodopsin in the dark-adapted and light-adapted state at an unprecedented resolution of 1.3 and 1.1 Å, respectively. This brings a wealth of novel molecular details, that are interpreted with help of an impressive array of biophysical techniques and in-silico simulations. All-in-all a ground-breaking manuscript. Interesting novel data are also presented with respect to posttranslational modifications of Archaerhodopsin-3, but the most essential message of course comprises the structural alterations recognized within and between the DA and LA states.

Authors' response: We thank the reviewer for these comments.

However, a major concern is that all data have been obtained under conditions (monomeric state in a very restrictive crystalline environment) that are far off from the trimeric state in the native claret membrane. The authors care-free translate their observations into a native perspective. In my view it is essential that the authors add a section clearly discussing and as much as possible substantiating to what extent their data and interpretation can be translated to the native membrane environment.

Authors' response: We agree with the reviewer that it would have been preferable to crystallise the protein as a trimer, however despite numerous attempts, we were not able to do so. We would note that the crystal structure for dark-adapted bacteriorhodopsin (PDB:1X0S) is also a monomer in the unit cell (see Crystal structure of the 13-cis isomer of bacteriorhodopsin in the dark-adapted state. Nishikawa, T., Murakami, M., Kouyama, T. (2005) *J Mol Biol* 352: 319-328). Following the reviewer's suggestion, we have discussed these issues on Page 17.

Major comments:

1. With an about 2:1 occupancy of the 13-cis and all-trans chromophore in the DA state (page 6, lines 4-6) I would expect a free energy difference for the all-trans <-> 13-cis equilibrium of about – 0.4 kcal/mol. This is much smaller than the calculated free energy difference (– 1.9 kcal/mol; page 8, top). Hence, the presented energy differences should only be considered as quite rough indicators for the all-trans/13-cis equilibrium.

Authors' response: We wrote in the manuscript that the computational results were consistent with the experimental findings, meaning the trends were correctly reproduced. Following the reviewer's recommendation, we have added a clarification on page 8.

2. It is suggested (page 8, lines 7-9), that in the LA state thermal all-*trans* to 13-*cis* isomerization does not occur because of the calculated higher activation energy compared to the DA state. However, this is not correct in my opinion. The free energy difference calculated for the all-trans/13-cis equilibrium in the LA state of about 10.9 kcal/mol is so high, that even if the real energy difference would be only half this value, the 13-*cis* population in this equilibrium would be negligible, irrespective of the height of the activation barrier.

Authors' response: We agree with the reviewer and have changed the manuscript accordingly (page 8).

3. Has it been taken into account that the thermal all-trans to 13-cis isomerization in the DA state comprises a reaction differing from the light-dependent one in the LA state? The first one is a 13-trans,15-anti to 13-cis,15-syn transition (the SB proton should remain in roughly the same position), while the latter is a 13-trans,15-anti to 13-cis,15-anti transition (the SB proton can then be transferred to the proton-accepting residue).

Authors' response: Yes, the differences between the various starting/product states of the retinal have been taken into account.

4. The authors imply (page 7, lines 4-7 from bottom) that upon light activation of the DA state only the all-trans population enters the photocycle with active proton pumping, while the depleting all-trans population is then repopulated via the thermal equilibrium with the 13-cis state. Considering the quite high activation barrier calculated for this equilibrium (17 kcal/mol; page 8, line 7), the rate of this thermal repopulation is relatively low. It is questionable whether in this way within two minutes the 13-cis state can be fully converted into all-trans, as can be clearly concluded from analysis of the LA crystals. To me a more realistic view is that the 13-cis population (13-cis,15-syn chromophore) is also excited by illumination, generating a 13-trans, 15-syn state, which thermally rapidly relaxes to the 13-trans,15-anti state.

Authors' response: We are grateful to the reviewer for this suggestion and have added this proposed mechanism to the discussion on Page 7.

5. Do the calculated free energy differences (page 8, top half) and C13=C14 dihedral-dependent energy levels (Fig. 3) pertain to the major or the minor trans state in the LA case? Are there significant differences in these aspects between these two trans states ?

Authors' response: The free energy profile of the LA state in Fig. 3 is based on both the major and minor trans states, as mentioned in the caption of Fig. 3: "Each point on the curve is generated from two independent 0.5 ns QM(SCC-DFTB)/MM MD trajectories, initiated from two separated equilibrated starting structures." We have added a sentence to the methodology to clarify this (p. 16, "Computation of potentials of mean force").

There were no significant differences between the free energy profiles derived only from either the major or the minor trans state and the one derived from the combination. To elucidate this, we have added to the supplementary information a graph (Supplementary Fig. 7) showing the free energy profiles derived only from the major and the minor trans states as well as the combined profile.

6. A major concern is that apparently during crystallization the native trimer organization is broken up into monomers that generate the crystal unit cell. For several microbial rhodopsins it has been reported, that the pump function may not be severely affected by monomerization, but biochemical and biophysical properties like the thermal stability and absorbance maximum are affected (e.g. Brouillette et al. (1989) *Proteins* 5(1): 38-46; Ranaghan et al. (2011) *JAmChemSoc* 133(45): 18318-18327; Tsukamoto et al. (2014) *JPhysChemB* 118(43): 12383-12394; Hussain et al. (2015) *JMolBiol* 427(6): 1278-1290; Kao et al. (2019) *JPhysChemB* 123(9): 2032-2039; Idso et al. (2019) *JPhysChemB* 123(19): 4180-4192; Ganapathy et al. (2020) *BBA* 1862: 183113). This implies also subtle structural rearrangements in the protein including the chromophore. Hence, it is conceivable that the water structure and distribution in the protein and around the SB region is perturbed upon monomerization. This aspect needs proper discussion, and the authors should present evidence that the structural differences observed between the all-trans and 13-cis DA states are not, or at the most only partially due to monomerization.

Authors' response: We have included a response to these points in the discussion section (page 17). We have made clear that the differences in the absorption spectra between trimeric and monomeric forms is mostly caused by the loss of exciton coupling between chromophores (present in the former and absent in the latter), rather than by structural rearrangements taking place in the retinal binding site. (See Pescitelli and Woody 2012 *J. Phys Chem. B* 116 23 6751-63; Fujimoto 2010 *J. Chem. Phys.* 133 124101; Hasselbacher *et al.* 1988 27 2540-2546.)

7. A similar concern pertains to the LA state, but there an additional caveat arises in the non-native environment in which the LA state was generated. The protein-protein and protein-lipid contacts in a monomeric crystal of course differ dramatically from the native trimeric organization in the claret membrane and present quite unnatural boundary conditions and dynamics. Hence, I consider it essential that the authors also for the LA state discuss to what extent this can obstruct the translation of their interpretations to the native trimeric organization. For instance, can the restricted dynamics in the crystalline state, together with the excess heat produced during excitation, which cannot be properly absorbed by the protein, result in artefactual generation of two all-trans states?

Authors' response: Again, we have included these points about monomer-trimer differences in the discussion section.

Minor comments:

8. The authors claim that AR3 generates a much faster photocycle and larger current than bR (page 3, line 8 from bottom; pages 10 and 11, bottom lines). However, there is contradictory evidence about the kinetics and pump activity of AR3 versus bR. For instance, Vogt *et al.* record about the same photocurrent for AR3 and bR (Vogt *et al.* (2013) *Biophys J* 105(9): 2055-2063).

Authors' response: We thank the reviewer for drawing our attention to this paper and have clarified that not all investigators have observed that AR3 produces a higher photocurrent.

9. Re: chromophore configuration in photocycle intermediates (page 4, lines 5-6): most photo-intermediates do not contain the 13-cis,15-syn configuration, but either the 13-cis,15-anti (early phase) or the 13-trans,15-anti (late phase) configuration (see comment 3 and Naito *et al.* (2019) *BiophysRev* 11(2): 167-181).

Authors' response: We agree with the reviewer and we have made this change.

10. AR3 was purified by isolating claret membrane vesicles. The Methods section should present the AR3 purity level in this preparation in % w/w of the protein content.

Authors' response: From an SDS-PAGE gel we estimate the content of AR3 in claret membrane protein total content being $78 \pm 2\%$. This is very similar to bacteriorhodopsin content in purple membrane (approx. 75%). We have added this data to the methods in the main paper.

11. The purified AR3 absorbance spectrum (Fig. S3b, black curve) has a strange asymmetrical shape. Has it been baseline overcorrected, or does it still contain some bacterioruberin?

Authors' response: We agree that some residual bacterioruberin is still present at low concentration. The spectrum has not been over-corrected. We have added an additional sentence to explain this point in the Supplementary Information.

12. Textual comments:

a. Not all microbial rhodopsins undergo dark-adaptation. Replace “are” (Abstract, line 2) by e.g. “contain”.

Authors' response: We have made a change to make this clear.

b. Replace “Although the wild-type protein is more usually classified as a light-driven transport protein, “(page 3, lines 16-17) by “Although the wild-type protein is classified as a light-driven proton pump, “.

Authors' response: We have made this change.

c. Insert “or as membrane voltage sensors” after “specific wavelengths” (page 3, line 9 from bottom). This application has become much more important than altering neural activity.

Authors' response: We have made this change.

d. All through the manuscript the chromophore of AR3 is named “retinal”. Indeed, the chromophore is generated by covalently binding retinal to a lysine amino group via a Schiff base, but the resulting entity is a derivative of retinal and is called a retinylidene moiety. Better replace in all these cases “retinal” by “chromophore” or “retinylidene element”.

Authors' response: We have clarified on pages 4 and 5 that the chromophore is in fact retinylidene, however we would note that most papers published in the field refer to retinal (especially the most recent publications involving time-resolved XFEL) and we feel that it is beneficial to follow this convention.

e. As stated at page 6, line 4, the modeled 13-cis to all-trans ratio in the DA state is 70 to 30%. Please correct the 3:1 ratio mentioned at page 14, line 3, accordingly.

Authors' response: We have made this change.

f. The sentence “Repeated bindingas water.” (page 15, lines 7-10) is not clear to me. Why would repeated binding have a larger effect on very rapidly reacting systems like H-bonded networks than a single binding event? In my view, this sentence does not add much to the statement in the previous sentence. Hence, I would recommend to omit the sentence starting with “Repeated binding...” or else clarify it better.

Authors' response: We have removed this sentence as recommended by the reviewer.

g. References 24 and 44 are the same, except that the volume number in 24 is incorrect.

Authors' response: We have corrected and combined the references.

h. The two W402 residue numbers in the legend of Fig. S7 should be corrected to W602.

Authors' response: We have corrected the figure legend.

Reviewer #4

Archaerhodopsins are light-activated photoreceptors that are prominent representatives of microbial rhodopsins, which have recently gained particular interest in the field of optogenetics where they enable control over the stimulation or inhibition of individual neurons. Archaerhodopsin-3 (AR3) catalyzes the light-driven transport of protons similar to its well-characterized homologue bacteriorhodopsin (BR). Juarez et al. have solved the crystal structures of AR3 in its ground (or light-adapted) state and in its dark-adapted state to outstanding resolutions of 1.1 Å and 1.3 Å. Based on these high-resolution structures the authors have analysed the differences in molecular interactions within and around the retinal binding pocket. Together with computational analyses their data provide detailed explanations for an observed thermodynamic equilibrium between the different chromophore conformations that is proposed as a molecular basis for photoreceptor desensitization.

The deposited crystal structures represent the first high-resolution structural data for AR3, giving detailed insight into the proposed proton translocation pathway. In addition, and probably more importantly, the analysis and comparison of the ground and desensitised dark-adapted state provide evidence for the molecular mechanism of photoreceptor desensitisation. These novel structural insights are valuable contributions toward a complete understanding of the mechanism of proton translocation in microbial rhodopsins and further facilitate the application of archaerhodopsins in optogenetics.

Overall, the submitted study is presented in a clear fashion and the data were thoroughly analyzed, leading to reasonable and convincing conclusions. The novelty of the findings and their significance for the microbial rhodopsin community and the field of optogenetics warrant publication of this work after some revisions.

Authors' response: We thank the reviewer for these comments.

The following issues should be addressed:

- The authors mention the faster photocycle kinetics of AR3 compared to its homologs and relate this phenomenon to distinct H-bonded networks around the proton release complex (PRC) that could provide multiple pathways for proton transfer. This is an interesting observation and hypothesis. However, from the text it is not obvious what underlying

molecular mechanism the authors hold responsible for this effect. Am I correct to assume that this proposition is based on the fact that proton uptake and release have been shown to be rate-limiting steps in the photocycle of microbial rhodopsins and that multiple exits could reduce this bottle-neck effect? I would suggest to elaborate this point a bit more.

Authors' response: We agree with the reviewer that proton uptake and release are rate-limiting steps and that multiple exit pathways could reduce the bottle-neck effect. We have added some additional discussion to clarify this point.

- On the same topic, a quick literature search revealed a study from Geng et al. (Photochemistry and photobiology, 2018) that discussed the importance of the highly conserved residues T164 and S165 (nomenclature in HeAR) in the faster turnover of ARs, which was not discussed in the presented manuscript. Do the novel AR3 structures provide any structural evidence that could corroborate this observation?

Authors' response: We thank the reviewer for bringing this paper to our attention. Thr164 and Ser165 are conserved in AR3 (the numbering is the same as for HeAR). We are unable to observe any differences between light- and dark-adapted AR3 for these two residues. We would observe that the study by Geng *et al.* was carried out using recombinant protein, expressed in *E. coli* BL21 (DE3) cells and that their kinetics measurements were performed in DDM detergent. It is possible that the differences observed between wild-type AR3 and the mutants are influenced by this environment. We therefore do not feel it necessary to comment on this paper in our current publication, but will keep the findings in mind during future work.

- As discussed, a similar distribution of all-*trans* and 13-*cis* retinal in the dark-adapted state is observed in BR. For the latter, isomeric composition has been found to be dependent, among other important factors, on the oligomeric state of the protein. Significantly altered rates for light-dark adaptation were observed for monomeric BR compared to the oligomer indicating certain regulatory interactions between neighboring protomers (Dencher et al., Biochemistry, 1983). Cross-protomer interactions that can directly affect the photocycle have also been demonstrated in the crystal structure of the blue-light absorbing proteorhodopsin (Ran et al., Acta Crystallographica Section D, 2013). Since the AR3 crystal structures was solved in the monomeric state such potential interactions that could occur in the native membrane environment cannot be observed. It would be interesting to investigate this possibility by comparing or combining the structural data of the dark- and light-adapted state from AR3 with the information from the trimeric structure of AR2 for example.

Authors' response: Unfortunately, we are unable to directly model an AR3 trimer using the AR2 structure as a guide. This is because in our AR3 structures, the apparent bacterioruberin binding site is occluded approximately half-way along its length by the Phe150 sidechain (see the discussion on page 14). We feel that since we are unable to fit the second chromophore into its predicted binding site, there would be little to be gained by attempting a modelling exercise without it. (As discussed on page 14, it is not clear whether the position of Phe150 is an artefact of the crystallisation process.) We have included a discussion of the effect of monomerization in the discussion section (page 17).

- The omega loop in AR3 seems to coordinate a sodium ion. Is this coordination required for the stabilization of the specific loop conformation, which in turn possibly enables binding of bacterioruberin? Interestingly, in the AR2 structure at 1.8Å (3WQJ) a water molecule was modeled in the same position.

Authors' response: We have reanalysed our equilibrium molecular dynamics simulations and, over a 300 ns timescale, we see no significant change in the conformation of the omega loop. We do however observe the Na⁺ ion exchanging with several times and therefore conclude that the stability of the loop is not dependent on the presence of the ion. We have therefore added three additional sentences to the discussion of this section to clarify the role of Na⁺ in this region.

- The AFM topograph in Supplementary Figure 4 is of low quality and misleading (probably recorded with an asymmetric tip). Please provide a topograph of higher quality and only display a region of the image with lattice lines (currently, the structure of the topograph is not easily discernible).

Authors' response: We have succeeded in improving the resolution of AR3 membranes by AFM. We now provide a new figure showing a 200 nm scan with an inset with higher magnification in which the AR3 trimeric organization can be observed. The lattice of this image is indicated in the Supplementary Table 2.

- For Supplementary Table 2, please use values (a, b, alpha) from images recorded from different adsorption of 2D crystals on mica. It could be that the used piezo scanner has a creep (i.e., in slow scan direction) – please check if AFM scanner is well calibrated (please indicate used scanner in Materials and Methods). Alternatively, the authors might record images of negatively stained 2D crystals by electron microscopy, and calculate and analyse power spectra to prove that vectors a and b are indeed different. Finally, please check numbers, e.g., of a and b (vectors) in Supplementary Table 2 and legend to Supplementary Figure 4 – the rounding is not consistent.

Authors' response: We have now imaged AR3 lattice at higher resolution and the new parameters of the lattice are indicated in the new Supplementary Table 2 (a=b=7.75 nm, alpha angle=64°), close to the reference value of bacteriorhodopsin.

- The abbreviation AFM is not used in manuscript, please remove from “Abbreviation List”.

Authors' response: We have removed the abbreviation.

REVIEWERS' COMMENTS

Reviewer #1 (Remarks to the Author):

The authors report two new X-ray structures for the desensitized dark-adapted state of archeorhodopsin-3 as well as the light-adapted state. The major findings are that with the very high resolution X-ray structures they are able to clearly establish the electron density due to the mixture of 13-cis and all-trans states in dark adapted archeorhodopsin-3, as well as the all-trans isomer in the light-adapted state. By introducing QM/M simulations they are furthermore able to suggest that the mixture of isomers in the dark-adapted state is in qualitative agreement with to the equilibrium distribution, which differs from the light-adapted state. Additional insights pertain to the pentameric cluster of water molecules that is implicated in the proton transfer mechanism, and water movements. (See forthcoming issue of *Angewandte Chemie* for a highly relevant paper.) All of these aspects are confirmatory of what is known for other microbial rhodopsins, yet add significantly to the corpus of knowledge of this important class of membrane proteins.

The paper is very well written and the figures are illustrative of the major findings, with comprehensive supplemental information that substantiates and adds to the body of the main paper. The article shows good scientific clarity and with the significance of the findings it is appropriate for publication in *Nature Communications*. The authors have made a strong effort to respond to the comments and criticisms of all the referees including my own. I am satisfied that in its revised form it meets the high standards expected for *Nature Communications*.

-- Michael F. Brown

Reviewer #2 (Remarks to the Author):

In my first reviewing, I was not able to understand insights derived from the structural data and submitted negative comments. Now I understand the insights after reading the authors' reply and the comments by other reviewers. I also clearly see that the crystal data are well and carefully analyzed after reading the authors' responses. Accordingly, I recommend the publication of this work in the present manuscript.

Reviewer #3 (Remarks to the Author):

Very satisfactorily, the authors have agreed with most of my comments and revised the manuscript accordingly. I have only two items left for final consideration.

Re: Supplementary Figure 4; The legend mentions "scan of AR3 2D crystals", while the Results section mentions "patches of the claret membrane". This is a bit confusing. I would recommend to change the legend text into "scan of AR3 2D crystalline arrangements in the claret membrane" or the-like.

Re: my original comment 6 concerning the potential artifacts due to monomerization of AR3 during crystal formation. I find the explanation by the authors too restricted. The exciton coupling in the trimer is largely responsible for the CD activity, and loss of this may contribute to small spectral changes in the monomer. However, for instance the protonation state of the counterion will govern larger spectral changes and can definitely be manipulated by the micro-environment of the protein, including oligomerization, type of lipids, solubilization etc. More importantly, monomerization must have structural implications, since the interaction pattern of the protein will change considerably. This can affect very "soft" structural elements like rotamer distribution, proton wires, H-bonded network distribution etc. All of this may perturb the physiological profile of the protein. I would recommend that the authors explain this a bit more explicit in the second last section of the discussion, next to the

potential deficits of the crystalline state.

Reviewer #4 (Remarks to the Author):

The manuscript by Juarez et al. presents novel and significant findings regarding the molecular mechanism of photoreceptor desensitization, which entails important consequences for future research on microbial rhodopsins, as well as their application in optogenetics. This was demonstrated by solving and analysing the light- and dark-adapted structures of Archeorhodopsin-3 at an unprecedented resolution for membrane proteins. The authors have satisfactorily addressed my - and in my opinion - the issues raised by the other reviewers, improving the quality of the final manuscript even more. Overall, the quality and significance of the presented work warrants publication in Nature Communications.

Response to Reviewers' Comments 2

We thank the reviewers for their comments, recommendations and suggestions. In addition to the points noted, we have corrected some minor typographical errors

Reviewer #1:

The authors report two new X-ray structures for the desensitized dark-adapted state of archeorhodopsin-3 as well as the light-adapted state. The major findings are that with the very high resolution X-ray structures they are able to clearly establish the electron density due to the mixture of 13-cis and all-trans states in dark adapted archeorhodopsin-3, as well as the all-trans isomer in the light-adapted state. By introducing QM/M simulations they are furthermore able to suggest that the mixture of isomers in the dark-adapted state is in qualitative agreement with to the equilibrium distribution, which differs from the light-adapted state. Additional insights pertain to the pentameric cluster of water molecules that is implicated in the proton transfer mechanism, and water movements. (See forthcoming issue of *Angewandte Chemie* for a highly relevant paper.) All of these aspects are confirmatory of what is known for other microbial rhodopsins, yet add significantly to the corpus of knowledge of this important class of membrane proteins.

The paper is very well written and the figures are illustrative of the major findings, with comprehensive supplemental information that substantiates and adds to the body of the main paper. The article shows good scientific clarity and with the significance of the findings it is appropriate for publication in *Nature Communications*. The authors have made a strong effort to respond to the comments and criticisms of all the referees including my own. I am satisfied that in its revised form it meets the high standards expected for *Nature Communications*.

Authors' response: We thank the reviewer for these comments.

Reviewer #2:

In my first reviewing, I was not able to understand insights derived from the structural data and submitted negative comments. Now I understand the insights after reading the authors' reply and the comments by other reviewers. I also clearly see that the crystal data are well and carefully analyzed after reading the authors' responses. Accordingly, I recommend the publication of this work in the present manuscript.

Authors' response: We thank the reviewer for these comments.

Reviewer #3 (Remarks to the Author):

Very satisfactorily, the authors have agreed with most of my comments and revised the manuscript accordingly. I have only two items left for final consideration.

Re: Supplementary Figure 4; The legend mentions "scan of AR3 2D crystals", while the Results section mentions "patches of the claret membrane". This is a bit confusing. I would recommend to change the legend text into "scan of AR3 2D crystalline arrangements in the claret membrane" or the-like.

Authors' response: We have amended the text to read: "AFM height image of a 200 nm scan of an AR3 2D crystalline array in the claret membrane".

Re: my original comment 6 concerning the potential artifacts due to monomerization of AR3 during crystal formation. I find the explanation by the authors too restricted. The exciton coupling in the trimer is largely responsible for the CD activity, and loss of this may contribute to small spectral changes in the monomer. However, for instance the protonation state of the counterion will govern larger spectral changes and can definitely be manipulated by the micro-environment of the protein, including oligomerization, type of lipids, solubilization etc. More importantly, monomerization must have structural implications, since the interaction pattern of the protein will change considerably. This can affect very "soft" structural elements like rotamer distribution, proton wires, H-bonded network distribution etc. All of this may perturb the physiological profile of the protein. I would recommend that the authors explain this a bit more explicit in the second last section of the discussion, next to the potential deficits of the crystalline state.

Authors' response: We have added the following text to the discussion:

"The dissociation of AR3 oligomers will inevitably alter the interactions between the individual molecules and their environment. This may, in turn, perturb the behavior of dynamic structural elements within the interior of the protein, including amino acid side chains (thereby influencing their protonation state), water molecules and the chromophore itself. It is therefore essential that structures of dark-adapted microbial rhodopsins, crystallised as trimers, be obtained in order to determine the influence of oligomerisation state on the mechanisms of receptor desensitization and resensitization."

Reviewer #4

The manuscript by Juarez et al. presents novel and significant findings regarding the molecular mechanism of photoreceptor desensitization, which entails important consequences for future research on microbial rhodopsins, as well as their application in optogenetics. This was demonstrated by solving and analysing the light- and dark-adapted structures of Archeorhodopsin-3 at an unprecedented resolution for membrane proteins. The authors have satisfactorily addressed my - and in my opinion - the issues raised by the other reviewers, improving the quality of the final manuscript even more. Overall, the quality and significance of the presented work warrants publication in Nature Communications.

Authors' response: We thank the reviewer for these comments.